# COVID-19's impact on hospital stays, mortality, and readmissions for poverty-related diseases, noncommunicable diseases, and injury groups in Thailand

Satiti Palupi[1,2], Kyaw Ko Ko Htet[2], Vorthunju Nakhonsri[3], Chumpol Ngamphiw[3], Peerapat Khunkham[3], Sanya Vasoppakarn[4], Narumol Atthakul[4], Sissades Tongsima[3], Chantisa Keeratipusana[3], Watcharapot Janpoung[3], Virasakdi Chongsuvivatwong[2]*

1 Department of Communicable Disease, East Java Provincial Health Office, Surabaya, Indonesia,
2 Department of Epidemiology, Faculty of Medicine, Prince of Songkla University, Hat Yai, Thailand,
3 National Biobank of Thailand, National Science and Technology Development Agency, Khlong Luang, Pathum Thani, Thailand, 4 Bureau of Service Quality Development, National Health Security Office, Lak Si, Bangkok, Thailand

* cvirasak@medicine.psu.ac.th

**Data Availability Statement:** All data cannot be shared with the public as it contains sensitive, individual data. Please contact the Thai Health

## Abstract

### Aims

This study aims to compare the trends in the quality of hospital care for WHO's three disease groups pre-, during, and post-COVID-19 pandemic peak in Thailand.

### Methods

The study utilized existing hospital admission data from the Thai Health Information Portal (THIP) database, covering the period from 2017 to 2022. We categorized WHO's three disease groups: poverty-related, noncommunicable, and injury groups using the International Classification of Diseases (ICD)—10 of initial admission of patients, and we analyzed three major outcomes: prolonged ($\geq$ 90th percentile) length of stay (LOS), hospital mortality, and readmission pre-, during, and post-COVID-19 pandemic peak. Relative weight (RW) of hospital reimbursements was used as a surrogate measure of the severity of the diseases.

### Results

The average prolonged LOS of patients with poverty disease pre-, during, and post-COVID-19 pandemic peak were 7.1%, 10.8%, 9.05%, respectively. Respective hospital mortality rates were 5.02%, 6.22%, 6.05% and readmission were 6.98/1,000, 6.16/1,000, 5.43/1,000, respectively. For non-communicable diseases, the respective proportions in the prolonged LOS were 9.0%, 9.12%, and 7.58%, with respective hospital mortality being 10.65%, 8.86%, 6.62%, and readmissions were 17.79/1,000, 13.94/1,000, 13.19/1,000, respectively. The respective prolonged LOS for injuries were 8.75%, 8.55%, 8.25%. Meanwhile, respective hospital mortality were 4.95%, 4.05%, 3.20%, and readmissions were 1.99/1,000, 1.60/1,000, 1.48/1,000, respectively. The RW analysis reveals diverse impacts on resource

Information Portal (THIP) gate keeper with data access requests: Mr. Nipon Rattankom, Department of Epidemiology, Prince of Songkla University Hatyai, Thailand, nipon.rdh@gmail.com. The Ethics Committee may be contacted at medpsu.ec@gmail.com.

**Funding:** Publication of the manuscript was supported by Fogarty International Center and the National Institute of Allergy and Infectious Diseases of the National Institutes of Health on the project "TB/MDR-TB Research Capacity Building in low and middle-income countries in Southeast Asia" under Awarded Number D43TW009522. The funders had no role in study design, data collection and analysis, decision to publish, or preparation of the manuscript.

**Competing interests:** The authors have declared that no competing interests exist.

utilization and costs. Most poverty-related and noncommunicable diseases indicate increased resource requirements and associated costs, except for HIV/AIDS and diabetes mellitus, showing mixed trends. In injuries, road traffic accidents consistently decrease resource needs and costs, but suicide cases show mixed trends.

## Conclusions

COVID-19 had a more serious impact, especially prolonged LOS and hospital mortality for poverty-related diseases more than noncommunicable diseases and injuries.

## Introduction

On March 11, 2020, the World Health Organization (WHO) officially declared the COVID-19 pandemic as a worldwide crisis. This unprecedented event significantly affected various aspects of human life, particularly in the realm of healthcare. Essential health services were disrupted in 90% of countries across the world, with disruptions including resource constraints, reductions in elective procedures, and delays in essential care, as highlighted by the WHO [1–3]. These disruptions had profound implications for the quality of healthcare people received during the pandemic, as non-COVID-19 patients experienced a decreased quality of hospital care or outcomes [4].

The quality of hospital care is the degree of value attributed to a hospital, assessed through multiple measurements [5]. It is gauged by the effectiveness with which health services implement the most recent evidence-based professional knowledge and practices [6]. Three outcome measures commonly used in the USA, Netherlands, United Kingdom, Italy, Belgium, and Australia to evaluate the quality of care in hospitals are prolonged length of stay (LOS), hospital mortality, and readmission [7].

The COVID-19 pandemic also has significantly impacted Thailand's health service, as it has in many other countries worldwide. A prior study revealed that health facilities in Bangkok had to cope with a particularly high number of COVID-19 cases during the pandemic, which might affect the quality of hospital care [8]. This highlights the importance of health service resilience to ensure the continuation of health facility services after the pandemic.

As of October 20, 2023, the Thai government has reported a total of 4,757,728 confirmed cases of COVID-19 [9]. Thailand's first wave of COVID-19 in March 2020, peaked on March 22, 2020. Subsequently, on March 26, 2020, the Thai government declared a state of emergency in response [10]. On October 1, 2022, the Thai government announced the end state of COVID-19 as an emergency disease or dangerous communicable disease into a communicable disease under surveillance [11].

To comprehensively understand the pandemic's impact on healthcare quality, this study aims to compare trends in hospital care quality for WHO's three disease groups pre-, during, and post-COVID-19 pandemic peak in Thailand.

## Materials and methods

### Study design and data source

The study design involved a series of analyses of existing data in the Thai Health Information Portal (THIP) database from 2017 to 2022.

THIP is a cooperative data warehouse project managed jointly by the National Health Security Office (NHSO) of Thailand, Prince of Songkla University, and the National Science and Technology Development Agency (NSTDA) of Thailand. The project retrieved claimed data from patient files at NHSO and NSTDA stores and saved the data to the user. PSU acts as the gatekeeper [12]. The information is used to support education and research. All records are de-identified.

In this study, THIP data was accessed between September 5, 2023, and December 16, 2023. The Institutional Ethics Committee of the Faculty of Medicine at Prince of Songkla University in Hat Yai, Thailand, granted ethical approval under reference number REC 66-485-18-1. Prof. Boonsin Tangtrakulwanich, Chairman of the Human Research Ethics Committee, Faculty of Medicine, Prince of Songkla University, approved the study.

The variables we used were encrypted personal ID, sex, age, admission date, discharge date, primary diagnosis (based on the International Classification of Diseases (ICD-10)), discharge status, and Relative Weight (RW) of DRG (diagnosis-related groups).

The diagnosis-related groups (DRG) prospective payment system was introduced under the Universal Health Coverage (UHC) scheme in Thailand to standardize reimbursement payments for inpatient hospital admissions, including those for cancer patients. This system categorizes diseases with similar characteristics into diagnostic and treatment groups to ensure uniformity in clinical processes and resource consumption costs. The relative weights (RW) of each DRG were determined based on existing hospital charges and health insurance reimbursement [13]. An elevated RW indicates increased resource requirements for treating a patient, culminating in higher medical costs [14]. Since the hospital resources used for COVID-19 diseases might vary over the study period, in comparison to the outcome, we used RW to make sure the effects of the period are comparable.

## Study population and time period

The study population was all patients admitted to Thailand hospitals between 2017 and 2022.

## Diseases groups

In this study, to assess the quality of hospital care for WHO's three disease groups, we categorized various diseases into three major groups: poverty-related diseases, noncommunicable diseases, and injuries [15, 16].

In this study, we used diarrhea, HIV/AIDS, influenza, pneumonia, and tuberculosis (TB), which was a highly spread infectious disease and was also a poverty-related disease as a representative of the poverty-related diseases group [15, 17, 18]. The noncommunicable disease group, we used cancer, chronic obstructive pulmonary disease (COPD), ischemic heart disease, hypertensive disease, stroke, and diabetes mellitus as representatives of the noncommunicable diseases group [15, 19], which was common noncommunicable diseases in the world and also associated with increased risk of complications and mortality from COVID-19 [19] and we explored road traffic accidents and suicide as representatives of unintentional injuries and intentional injuries in the injuries group [15, 20]. Period pre-, during and post-COVID-19 pandemic.

Fig 1 illustrates the trend of COVID-19 cases on a logarithmic scale. The study timeline was categorized into three phases: the pre-COVID-19 pandemic peak period (prior to March 2020) encompassing a time without implemented public health measures, the during-COVID-19 pandemic peak period (up to April 2022), marked by diverse public responses such as national lockdowns and hospital use restrictions; and the post-COVID-19 pandemic peak period, denoting a phase after the lifting of all implemented measures.

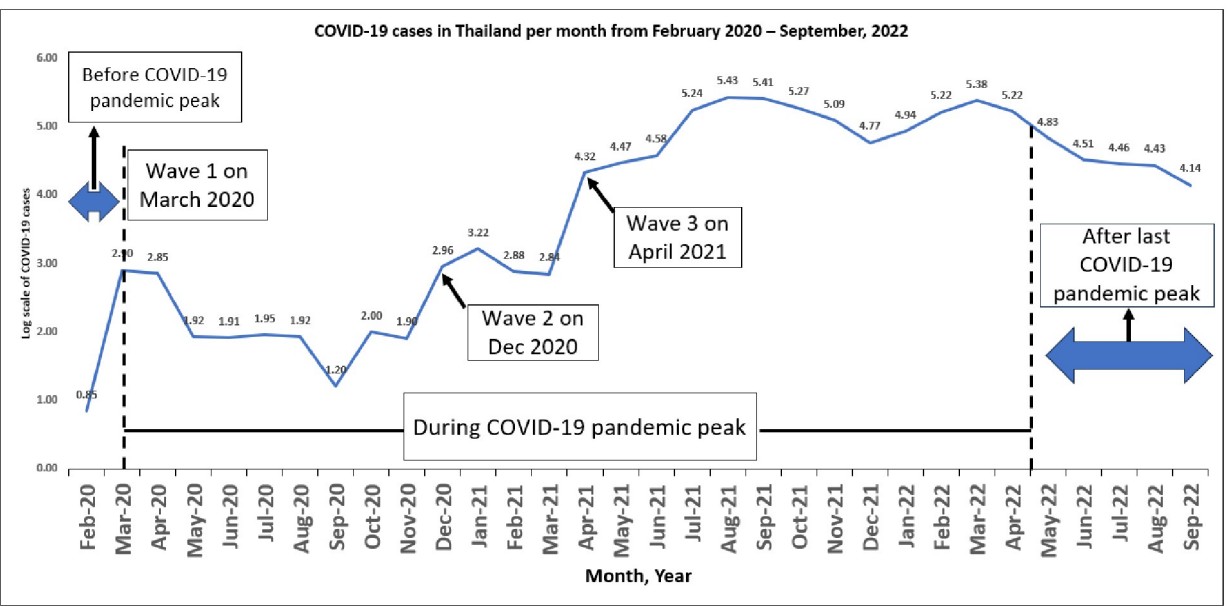

**Fig 1. Fluctuations in COVID-19 cases in Thailand from time to time (February 22, 2020–September 30, 2022).**

## ICD-10 grouping

Table 1 lists disease codes grouped into three categories.

## Outcomes

Length of stay (LOS) was the length of time elapsed between a patient's hospital admittance and discharge. The average length of stay (LOS) was calculate for individual hospitalization, greater and equal to the 90th percentile for the specific diagnosis or procedure group (upper bound of outlier length of stay) is considered as prolonged LOS [7, 21]. Hospital mortality is defined as death from any cause in the hospitalized population, recorded in by hospital discharge status [22, 23]. Finally, readmission is defined as unplanned subsequence acute care

**Table 1. ICD-10 codes of the diseases.**

| Group | Diseases | Subgroup | ICD-10 codes |
|---|---|---|---|
| **Poverty-related diseases** | Diarrhea | | A00, A01, A03, A04, A06-A09 |
| | HIV/AIDS | | B20–B24 |
| | Influenza | Lower respiratory infection | J09–J11 |
| | Pneumonia | Lower respiratory infection | J12-J18 |
| | Tuberculosis (TB) | | A15–A19 |
| **Noncommunicable disease** | Cancer | | C00–C97 |
| | Chronic Obstructive Pulmonary Disease (COPD) | | J40–J47 |
| | Ischemic heart disease | Chronic Vascular Disease (CVD) | I20–I25 |
| | Hypertensive disease | Chronic Vascular Disease (CVD) | I10–I13 |
| | Stroke | Chronic Vascular Disease (CVD) | I63, I65, I66 |
| | Diabetes mellitus (DM) | | E10–E14 |
| **Injuries** | Road traffic accident | | V01–V99 |
| | Suicide | | X60–X84 |

inpatient admission of the same patient within 72 hours of discharge of the initial inpatient acute care admission [24, 25].

## Statistical analysis

The data will be analyzed by using R version 3.1.1 (R Computing, Vienna, Austria).

Main outcome variables include whether the patients had the specific outcome at discharge time. Main independent variable included the period (pre-COVID-19 pandemic peak, during COVID-19 pandemic peak and post-COVID-19 pandemic peak). Covariates used for adjustment include age, sex, and RW.

Univariate analyses were used to scan the association between the independent and the outcome variables. Then we conducted logistic regression to assess the differences in COVID-19 pandemic peak period (pre-, during, and post-) in relation to prolonged length of stay, hospital mortality, and readmission in each disease with adjustment of sex, age, and RW.

## Results

### General description

From 2017 to 2022, altogether, 4,819,921 patients were admitted from 2017, to 2022, with ICD-10 codes of interest in primary diagnoses. Table 2 shows data on sex and age distribution across different time periods (pre-, during, and post-COVID-19 pandemic peak). In this study, 54.75% patients were males. The mean age was higher during COVID-19 pandemic peak than other periods.

### Prolonged length of stay

Table 3 shows the 90th percentile of length of stay of various diseases in descending order, note that TB ranked the second only CVD.

**Table 2. Baseline characteristics of admitted patients in hospital.**

| | Period of COVID-19 pandemic peak | | | |
|---|---|---|---|---|
| **Variables** | **Pre-** | **During** | **Post-** | **Overall** |
| **Sex** | (N = 2,376,649) | (N = 1,418,160) | (N = 273,976) | (N = 4,068,790) |
| Female | 1,021,471 | 603,555 | 118,297 | 1,743,323 |
| Male | 1,355,178 | 814,605 | 155,679 | 2,325,462 |
| **Age (year)** | | | | |
| Mean (SD) | 58.8 (± 20.5) | 59.1 (±19.6) | 59.0 (±19.7) | 58.9 (± 20.2) |
| Median (min, max) | 63.0 (0, 117) | 63.0 (0, 122) | 63.0 (0, 117) | 63.0 (0, 122) |
| **Poverty-related diseases group** | | | | |
| Diarrhea | 233,011 | 108,582 | 20,845 | 362,438 |
| HIV/AIDS | 10,878 | 5,502 | 1,040 | 17,420 |
| Lower respiratory Infection (LRI) | 972,797 | 745,374 | 134,790 | 1,852,961 |
| Tuberculosis (TB) | 101,763 | 59,920 | 10,613 | 172,296 |
| **Non-communicable diseases group** | | | | |
| Cancer | 302,293 | 196,676 | 38,486 | 537,455 |
| Chronic Obstructive Pulmonary Disease (COPD) | 162,678 | 58,361 | 12,454 | 233,493 |
| Cardiovascular disease (CVD) | 631,170 | 388,910 | 75,595 | 1,095,675 |
| Diabetes mellitus (DM) | 42,932 | 32,798 | 6,440 | 82,170 |
| **Injuries group** | | | | |
| Road traffic accident | 224,446 | 152,025 | 26,478 | 402,949 |
| Suicide | 36,308 | 26,479 | 6,432 | 69,219 |

**Table 3. 90<sup>th</sup> percentile of length of stay of various diseases of interest.**

| Diseases | 90th percentile of length of stay (days) |
|---|---|
| **Poverty-related diseases group** | |
| Diarrhea | 5 |
| HIV/AIDS | 13 |
| LRI | 14 |
| TB | 17 |
| **Non-communicable diseases group** | |
| Cancer | 14 |
| COPD | 9 |
| CVD | 37 |
| DM | 5 |
| **Injuries group** | |
| Road traffic accident | 5 |
| Suicide | 5 |

## Poverty-related diseases group

Table 4 provides a comprehensive overview indicating a consistent rise in both the percentage and average percentage of prolonged LOS during and post-COVID-19 pandemic peak within the poverty-related diseases group. Concurrently, there was a consistent upward trend in the

**Table 4. Percentages of prolonged length of stay and hospital mortality, per thousand readmission, and relative weight of poverty-related diseases group between pre-, during and post-COVID-19 pandemic peak.**

| Outcomes of poverty-related diseases group | Period of COVID-19 pandemic peak | | | P value |
|---|---|---|---|---|
| | **Pre-** | **During** | **Post-** | |
| **Prolonged length of stay** | Percentage (%) | Percentage (%) | Percentage (%) | |
| Diarrhea | 7.8 | 9.8 | 10.2 | <0.001 |
| HIV/AIDS | 9.8 | 11.2 | 10.3 | 0.0155 |
| LRI | 0.5 | 10.7 | 5.2 | <0.001 |
| TB | 10.3 | 11.6 | 10.5 | <0.001 |
| **Average** | **7.1** | **10.8** | **9.05** | **<0.001** |
| **Hospital mortality** | Percentage (%) | Percentage (%) | Percentage (%) | |
| Diarrhea | 1.2 | 1.5 | 1.4 | <0.001 |
| HIV/AIDS | 7.8 | 8.6 | 7.5 | <0.001 |
| LRI | 3.5 | 5.6 | 6.2 | 0.4009 |
| TB | 7.6 | 9.2 | 9.1 | <0.001 |
| **Average** | **5.025** | **6.22** | **6.05** | **<0.001** |
| **Readmission** | Per thousand | Per thousand | Per thousand | |
| Diarrhea | 2.34 | 2.07 | 1.54 | 0.0072 |
| HIV/AIDS | 8.22 | 10.29 | 7.34 | 0.4319 |
| LRI | 4.70 | 2.09 | 2.85 | <0.001 |
| TB | 12.65 | 10.18 | 9.99 | <0.001 |
| **Average** | **6.98** | **6.16** | **5.43** | **<0.001** |
| **Relative weight (RW)** | $\bar{x} \pm SD$ | $\bar{x} \pm SD$ | $\bar{x} \pm SD$ | |
| Diarrhea | 0.426 ± 0.65 | 0.473 ± 0.744 | 0.471 ± 0.652 | <0.001 |
| HIV/AIDS | 1.244 ± 1.319 | 1.285 ± 1.448 | 1.227 ± 1.072 | 0.0753 |
| LRI | 1.016 ± 2.21 | 1.161 ± 2.139 | 1.359 ± 2.215 | <0.001 |
| TB | 1.624 ± 2.293 | 1.848 ± 2.485 | 1.853 ± 2.22 | <0.001 |

percentage of hospital mortality in this group, in line with an increase in the average percentage, except for HIV/AIDS, which displayed mixed trends. Conversely, most readmissions in poverty-related diseases decreased during and post-COVID-19 pandemic peak, aligning with the overall decrease in the average per thousand patient readmissions in this group. However, HIV/AIDS presented mixed trends in this context.

The relative weights (RW) for diarrhea, LRI, and TB show consistently increased during, and post- compared with pre-COVID-19 pandemic peak. These changes indicate an increase in resource requirements and average costs associated with severe illness of patients with these diseases.

The results of logistic regression in S1 Table, predicting prolonged length of stay, hospital mortality, and readmission of poverty-related diseases with adjustment for sex, age, and RW, were similar to the result in Table 4 (without the adjustment).

## Noncommunicable diseases group

Table 5 summarizes that the percentage of prolonged LOS during and post-COVID-19 pandemic peak consistently increased in COPD, decreased in cancer, and had mixed trends in CVD and DM compared to pre-COVID-19 pandemic peak. The average percentages of prolonged LOS also exhibited mixed trends. Concurrently, hospital mortality consistently decreased, along with readmissions during and post-COVID-19 pandemic peak, aligning with

**Table 5. Percentages of prolonged length of stay and hospital mortality, per thousand readmission, and relative weight of noncommunicable diseases group between pre-, during, and post-COVID-19 pandemic peak.**

| Outcomes of noncommunicable diseases group | Period of COVID-19 pandemic peak | | | P value |
|---|---|---|---|---|
| | Pre- | During | Post- | |
| **Prolonged length of stay** | Percentage (%) | Percentage (%) | Percentage (%) | |
| Cancer | 9.7 | 8.3 | 7.4 | <0.001 |
| COPD | 7.7 | 9.1 | 8.4 | <0.001 |
| CVD | 10.0 | 10.9 | 5.9 | 0.0079 |
| DM | 8.6 | 8.2 | 8.6 | 0.0181 |
| **Average** | **9.0** | **9.12** | **7.58** | **<0.001** |
| **Hospital mortality** | Percentage (%) | Percentage (%) | Percentage (%) | |
| Cancer | 19.8 | 16.2 | 11.2 | <0.001 |
| COPD | 8.3 | 6.8 | 5.5 | <0.001 |
| CVD | 8.5 | 8.3 | 7.3 | <0.001 |
| DM | 6.0 | 4.2 | 2.5 | <0.001 |
| **Average** | **10.65** | **8.87** | **6.62** | **<0.001** |
| **Readmission** | Per thousand | Per thousand | Per thousand | |
| Cancer | 23.77 | 18.15 | 15.92 | <0.001 |
| COPD | 33.54 | 24.95 | 25.16 | <0.001 |
| CVD | 8.75 | 8.15 | 7.16 | <0.001 |
| DM | 5.09 | 4.49 | 4.53 | 0.0408 |
| **Average** | **17.79** | **13.94** | **13.19** | **<0.001** |
| **Relative weight (RW)** | $\bar{x} \pm SD$ | $\bar{x} \pm SD$ | $\bar{x} \pm SD$ | |
| Cancer | 2.742 ± 2.336 | 2.805 ± 2.262 | 2.803 ± 2.17 | <0.001 |
| COPD | 1.396 ± 2.392 | 1.478 ± 2.4 | 1.536 ± 2.188 | <0.001 |
| CVD | 3.532 ± 4.265 | 3.835 ± 4.409 | 3.833 ± 4.284 | <0.001 |
| DM | 2.262 ± 3.248 | 2.345 ± 3.173 | 2.208 ± 2.653 | <0.001 |

the observed patterns in the average percentages of hospital mortality and readmissions per thousand in this group.

The relative weights for cancer, COPD, and CVD consistently increased during, and post-compared with pre-COVID-19 pandemic peak. These changes indicate an increase in resource requirements and average costs associated with severe illness of patients with these diseases. Meanwhile, DM shows mixed trends.

As shown in S2 Table, logistic regression with adjustment for sex, age, and RW had similar results to those in Table 5 (without the adjustment).

## Injuries group

Table 6 provides a summary of the percentage of prolonged length of stay (LOS) during and post-COVID-19 pandemic peak, revealing mixed trends within the injuries group when compared to the pre-COVID-19 pandemic peak. On the other hand, the average percentage of prolonged LOS demonstrated a decrease during and post-COVID-19 pandemic peak. Meanwhile, hospital mortality consistently decreased in cases of suicide during, and post-COVID-19 pandemic peak compared to the pre-COVID-19 pandemic peak period, along with a reduction in readmissions. This aligns with the observed patterns in the average percentage of hospital mortality and per thousand readmission in this context.

The relative weights for road traffic accidents show consistently decreased during and post-compared with pre-COVID-19 pandemic peak. These changes indicate a reduction in resource requirements and average costs associated with severe illness of patients with these diseases. Meanwhile, suicide shows mixed trends.

The S3 Table shows logistic regression results after adjustment for sex, age, and RW, which are similar to those in Table 6 (without the adjustment).

We summarize the direction of effect and consistent findings among three major disease groups with the quality of hospital care outcomes as shown in Table 7. Poverty-related diseases exhibit challenges, with consistently increased prolonged length of stay and hospital mortality.

**Table 6. Percentages of prolonged length of stay and hospital mortality, per thousand readmission, and relative weight of injuries group between pre-, during, and post-COVID-19 pandemic peak.**

| Outcomes of injuries group | Period of COVID-19 pandemic peak | | | P value |
|---|---|---|---|---|
| | Pre- | During | Post- | |
| **Prolonged length of stay** | Percentage (%) | Percentage (%) | Percentage (%) | |
| Road traffic accident | 8.6 | 8.2 | 8.6 | 0.0181 |
| Suicide | 8.9 | 8.9 | 7.9 | 0.0422 |
| Average | **8.75** | **8.55** | **8.25** | |
| **Hospital mortality** | Percentage (%) | Percentage (%) | Percentage (%) | |
| Road traffic accident | 3.9 | 3.9 | 3.9 | 0.2981 |
| Suicide | 6.0 | 4.2 | 2.5 | <0.001 |
| Average | **4.95** | **4.05** | **3.2** | |
| **Readmission** | Per thousand | Per thousand | Per thousand | |
| Road traffic accident | 2.13 | 1.96 | 2.37 | 0.856 |
| Suicide | 1.85 | 1.25 | 0.59 | 0.0045 |
| Average | **1.99** | **1.61** | **1.48** | |
| **Relative weight (RW)** | $\bar{x} \pm SD$ | $\bar{x} \pm SD$ | $\bar{x} \pm SD$ | |
| Road traffic accident | 2.932 ± 4.09 | 2.841 ± 3.844 | 2.727 ± 3.451 | <0.001 |
| Suicide | 0.512 ± 1.271 | 0.528 ± 1.317 | 0.476 ± 0.983 | 0.0079 |

**Table 7. Direction of effect and consistent findings among disease groups and the outcomes of quality of hospital care.**

| Groups | Poverty-related diseases | Noncommunicable diseases | Injuries |
|---|---|---|---|
| **The severity of illness as reflected by relative weight (RW) of hospital reimbursement** | Mostly increase consistently | Mostly increase consistently | Road traffic injuries RW decrease consistently. Suicide RW increased during COVID pandemic peak |
| **Outcomes** | | | |
| Prolonged length of stay | Increase consistently | Mixed trend | Mixed trend |
| Readmission | Mostly decrease consistently | Decrease consistently | Readmission of road traffic injuries did not significantly. |
| | | | Suicide decreases in readmission |
| Hospital mortality | Mostly increase consistently | Decrease consistently | Hospital mortality of road traffic injuries did not change significantly. |
| | | | Suicide decreases in hospital mortality |

Noncommunicable diseases present mixed trends in prolonged length of stay, but consistently show improvements with decreased hospital mortality and readmissions.

## Discussion

The severity of poverty-related diseases and noncommunicable diseases increased consistently. The prolonged hospital stay was observed mainly in cases of poverty-related diseases. Hospital mortality increased consistently in the diseases of poverty-related diseases but decreased consistently in noncommunicable diseases and suicide. Finally, readmission decreases consistently in nearly all disease groups.

The RW analysis highlights varying impacts on resource utilization and costs associated with severe illness across different diseases. An upward trend for most diseases in poverty-related diseases and noncommunicable diseases signaling increased resource requirements and associated costs linked to severe illness.

It might be due to the fact that the COVID-19 pandemic deepened the deprivation of low socioeconomic status (SES) groups [26, 27], which then led to more complications, such as the diseases of poverty. Furthermore, the COVID-19 pandemic might delay decisions and facilities of low SES groups in arriving at the health service timely. These phenomena were likely in chronic noncommunicable diseases, the diseases in which groups progress more slowly, and in road traffic accidents where prompt response by the rescue system was not much disrupted by the pandemic. The same explanation could be applied to prolonged length of stay.

Effects on hospital readmission were quite different from that on length of stay. In general, readmission is partially affected too early discharge [28], when the final diagnosis and/or treatment were not reached. The COVID-19 pandemic decreased the hospital's general caseload due to the lockdown process and people who were generally scared of contracting the infection. Reduction of the general caseload could reduce the need for early discharge of the patients to reduce hospital congestion reduction of early discharge, thus reducing the chance of readmission.

Across various disease groups, poverty-related diseases generally showed an increase in both prolonged LOS and hospital mortality during and post-pandemic periods. HIV/AIDS was the exception, which exhibited mixed trends. Conversely, readmissions for poverty-related diseases generally decreased during and post-COVID-19 pandemic peak. Similar trends were observed in noncommunicable diseases, where COPD demonstrated an increased prolonged LOS, cancer consistently decreased, and hospital mortality decreased across various diseases. In the injuries group, the study found mixed trends in the percentage of prolonged LOS, but

hospital mortality consistently decreased, particularly in cases of suicide, and readmissions declined.

Respiratory distress and gastrointestinal symptoms are common in many infectious diseases such as COVID-19, influenza, pneumonia, and even TB [29, 30]. To rule out COVID-19 transmission in the pandemic period, the hospitalization period needed to be extended, which led to the lengthening of LOS. Patients might have prolonged delays in arriving at the hospital to get service due to worrying about getting a COVID-19 infection from the hospital and inconvenience in transportation and the referral system. These delays would increase the clinical complication level, leading to extended LOS. The change of pattern in healthcare systems during the pandemic could result in delayed access to medical care, leading to more severe presentations of these conditions when patients finally seek treatment [31].

Lockdowns may limit people's ability to seek timely healthcare for various conditions, leading to more severe presentations when they eventually seek treatment. In addition, during the pandemic, delayed diagnosis of new cases often led to severe disease complications [32]. These new cases of complications are also probably accumulated in the post-pandemic period, particularly for poverty-related diseases such as TB, respiratory diseases, and HIV. Although delayed diagnosis of new cases can also occur in COPD and cancer patients, the slow progression of these new diseases means they are less likely to present as severe cases in the post-pandemic period. More severe cases often require longer hospital stays or hospital mortality. More severe cases often require longer hospital stays or hospital mortality [33–35].

Besides that, COVID-19 could cause severe respiratory complications, and individuals with pre-existing conditions, such as HIV, influenza, pneumonia, and TB, were more susceptible to these complications. However, the magnitude of these problems is not known.

People were worried about contracting COVID-19 while in a healthcare setting. Hospitals were perceived as places where the risk of exposure to the virus may be elevated [36]. According to several studies, many people in the USA and other countries have delayed or avoided medical care during the pandemic due to fear of COVID-19, financial difficulties, or lack of access [37–39]. This could lead to worse health outcomes for those with chronic or acute conditions [40, 41].

Prolonged LOS consistently increased during and post-COVID-19 pandemic peak for COPD, potentially due to the similarities in clinical presentation between COPD exacerbations and COVID-19 [42, 43]. This overlap in symptoms may have led to challenges in differentiating between the two conditions, potentially delaying appropriate treatment and contributing to longer hospital stays. Additionally, the pandemic may have strained healthcare resources, limiting the availability of specialized care for COPD patients and further contributing to prolonged LOS [44].

Based on previous studies, there are home chemotherapy programs in Thailand, which may be connected to adaptations in healthcare delivery during lockdowns [45]. The mention of a home chemotherapy program in Thailand suggests a shift towards more efficient and possibly remote care delivery, which could be a response to the limitations imposed by lockdown policies. Previous research also conducted in Thailand during the pandemic showed that DM patients could be resilient using cost-effective and standard digital technologies to provide a continuum of care for DM patients, and alternative services like mobile medical labs, medication delivery, and medical refills at drug stores can improve continuous monitoring of glycemic control and prescribed medication use [8]. Thailand also has had a universal health care program that eases the impact of DM since pre-COVID-19 pandemic. This has caused the Thai government to be well-prepared to handle DM during and post-COVID-19 pandemic [46, 47]. Reduced mortality was observed in patients with noncommunicable diseases during and post-COVID-19 pandemic peak. This may reflect success in the Thai health system. In

Thailand, health programs adopted care models to ensure the safety of care for people living with noncommunicable diseases during the pandemic [48].

The observed decrease in readmissions within three days for both poverty-related diseases and noncommunicable diseases, as well as for cases of suicide. These findings may be indicative of changes in healthcare utilization patterns, possibly influenced by pandemic-related factors such as altered healthcare-seeking behavior, increased awareness of preventive measures, or changes in healthcare delivery systems [49–51].

Poverty-related diseases have higher mortality rates during the pandemic due to increased exposure, vulnerability and barrier to health care of low socioeconomic status groups during the pandemic [52]. On the other hand, the decrease in hospital mortality rates in NCDs may be due to the increased focus on preventive measures and early detection of NCDs during the pandemic [53, 54].

The decrease in injuries hospital mortality in suicide may be due to the increased awareness of mental health issues during the pandemic. This study result was in line with the previous study in the USA that suicide deaths between 2019 and 2020 decreased by 3% overall [55]. In the lockdown period, although depression could become more common, people are confined together in the same premises. This may lessen the chance of an individual to make a commit suicide. Supportive relationships and mental health interventions played crucial roles in suicide prevention [56].

These findings provide a nuanced understanding of the pandemic's multifaceted impact on hospital care quality across different disease categories in the Thai healthcare system, offering valuable insights for future healthcare planning and improvements.

Furthermore, using relative weight (RW) as a surrogate measure of disease severity provides a novel perspective on the pandemic's economic impact on healthcare resource utilization and costs, adding a unique dimension to the existing literature.

This study has several limitations. First, it relies on hospital admission data, which may not fully capture the broader health outcomes of the entire population. Second, the data lacks information on disease severity, which could be a potential confounder in the analysis. Third, the use of relative weight (RW) as a surrogate measure of disease severity has its limitations, as it may not fully reflect the complexity and heterogeneity of individual cases. Finally, the study is observational. That limitation emphasizes the need for future research to adopt a more holistic approach to understanding the comprehensive implications of healthcare changes during and post-COVID-19, incorporating community-level health indicators and qualitative assessments alongside quantitative data. This research should investigate the long-term impact of the pandemic on hospital care quality, examine the effects on specific subpopulations, and explore health disparities, the role of healthcare infrastructure, and the effectiveness of public health interventions. Comparative studies across different countries could provide valuable insights for global pandemic preparedness and response.

The results of this study have several implications for healthcare policy and resource allocation in Thailand: prioritizing cases for vulnerable populations, enhancing healthcare infrastructure, optimizing resource allocation, and strengthening pandemic preparedness.

## Conclusions

COVID-19 had a more serious impact, especially prolonged LOS, and hospital mortality for poverty-related diseases more than noncommunicable diseases and injuries. More attention should therefore be given to improving care to the poor.

## Supporting information

**S1 Table. Logistic regression analysis of the association between prolonged length of stay, hospital mortality, and readmission of poverty-related diseases group and period of COVID-19 pandemic peak with adjustment for sex, age, and RW.**
(DOCX)

**S2 Table. Logistic regression analysis of the association between prolonged length of stay, hospital mortality, and readmission of noncommunicable diseases group and period of COVID-19 pandemic peak with adjustment for sex, age, and RW.**
(DOCX)

**S3 Table. Logistic regression analysis of the association between prolonged length of stay, hospital mortality, and readmission of injuries group and period of COVID-19 pandemic peak with adjustment for sex, age, and RW.**
(DOCX)

## Acknowledgments

This study was a part of a post-doctoral trainee of Satiti Palupi at the Department of Epidemiology, Faculty of Medicine, Prince of Songkla University, Hat Yai, Songkhla, Thailand. Thanks to the Fogarty International Center and the National Institute of Allergy and Infectious Diseases of the National Institutes of Health for their support of the project. The content is solely the responsibility of the authors and does not necessarily represent the official views of the National Institutes of Health. We also thank the Thai Health Information Portal (THIP) for providing the dataset.

## Author Contributions

**Conceptualization:** Satiti Palupi, Virasakdi Chongsuvivatwong.

**Data curation:** Satiti Palupi, Kyaw Ko Ko Htet, Virasakdi Chongsuvivatwong.

**Formal analysis:** Satiti Palupi, Kyaw Ko Ko Htet, Virasakdi Chongsuvivatwong.

**Funding acquisition:** Virasakdi Chongsuvivatwong.

**Investigation:** Satiti Palupi, Kyaw Ko Ko Htet.

**Resources:** Vorthunju Nakhonsri, Chumpol Ngamphiw, Peerapat Khunkham, Sanya Vasoppakarn, Narumol Atthakul, Sissades Tongsima, Chantisa Keeratipusana, Watcharapot Janpoung.

**Supervision:** Virasakdi Chongsuvivatwong.

**Visualization:** Satiti Palupi, Virasakdi Chongsuvivatwong.

**Writing – original draft:** Satiti Palupi, Virasakdi Chongsuvivatwong.

**Writing – review & editing:** Satiti Palupi, Kyaw Ko Ko Htet, Vorthunju Nakhonsri, Chumpol Ngamphiw, Peerapat Khunkham, Sanya Vasoppakarn, Narumol Atthakul, Sissades Tongsima, Chantisa Keeratipusana, Watcharapot Janpoung, Virasakdi Chongsuvivatwong.

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
