## [Decision Letter · Decision Letter 0]

15 Apr 2024

PONE-D-24-06438Quality of Hospital Care for WHO's Three Disease Groups Pre-, During, and Post-COVID-19 Pandemic Peak in ThailandPLOS ONE

Dear Dr. Chongsuvivatwong,

Thank you for submitting your manuscript to PLOS ONE. After careful consideration, we feel that it has merit but does not fully meet PLOS ONE’s publication criteria as it currently stands. Therefore, we invite you to submit a revised version of the manuscript that addresses the points raised during the review process.

We look forward to receiving your revised manuscript.

Kind regards,

Worradorn Phairuang, Ph.D.

Academic Editor

PLOS ONE

Journal Requirements:

"Research reported in this publication was supported by Fogarty International Center and the National Institute of Allergy and Infectious Diseases of the National Institutes of Health on the project “TB/MDR-TB Research Capacity Building in low and middle-income countries in Southeast Asia” under Awarded Number D43TW009522. "

" Thanks to the Fogarty International Center and the National Institute of Allergy and Infectious Diseases of the  National Institutes of Health on the project “TB/MDR-TB Research Capacity Building in low and middle-income countries in Southeast Asia” under Awarded Number D43TW009522, for the support."

"Research reported in this publication was supported by Fogarty International Center and the National Institute of Allergy and Infectious Diseases of the National Institutes of Health on the project “TB/MDR-TB Research Capacity Building in low and middle-income countries in Southeast Asia” under Awarded Number D43TW009522. "

Additional Editor Comments:

**ACADEMIC EDITOR: Major Revision**

Reviewers' comments:

Reviewer's Responses to Questions

**Comments to the Author**

1. Is the manuscript technically sound, and do the data support the conclusions?

Reviewer #1: Yes

Reviewer #2: Yes

Reviewer #3: Partly

2. Has the statistical analysis been performed appropriately and rigorously? 

Reviewer #1: Yes

Reviewer #2: Yes

Reviewer #3: No

3. Have the authors made all data underlying the findings in their manuscript fully available?

Reviewer #1: Yes

Reviewer #2: Yes

Reviewer #3: No

4. Is the manuscript presented in an intelligible fashion and written in standard English?

Reviewer #1: No

Reviewer #2: Yes

Reviewer #3: No

5. Review Comments to the Author

Reviewer #1: The study is not novel and similar papers were published before.

Reviewer #2: The article titled "Quality of Hospital Care for WHO's Three Disease Groups Pre-, During, and Post-COVID-19 Pandemic Peak in Thailand," submitted to the scientific journal Plos One, delves into the critical examination of hospital care quality across different disease groups during various phases of the COVID-19 pandemic. Through analyzing trends in hospital admission data from the Thai Health Information Portal (THIP) database, the study aims to shed light on how the pandemic has affected the quality of care for poverty-related, noncommunicable, and injury-related diseases in Thailand.

Positive Points:

Relevance and Timeliness: The study addresses a timely and significant issue, providing insights into how the COVID-19 pandemic has impacted the quality of hospital care across different disease groups. This research is particularly valuable given the ongoing global healthcare crisis and the need to understand its implications on healthcare delivery systems.

Comprehensive Methodology: The utilization of existing hospital admission data from the THIP database covering a substantial period from 2017 to 2022 lends credibility to the study. Categorizing disease groups using the International Classification of Diseases (ICD)-10 and analyzing outcomes such as prolonged length of stay (LOS), hospital mortality, and readmission pre-, during, and post-COVID-19 pandemic peak offers a comprehensive view of the situation.

Insightful Findings: The study's findings provide valuable insights into the impact of the pandemic on different disease groups. By examining metrics such as prolonged LOS, hospital mortality rates, and readmissions, the research highlights the varying degrees of impact across poverty-related, noncommunicable, and injury-related diseases, contributing to a deeper understanding of healthcare dynamics during the pandemic.

Opportunities for Improvement:

Clarity in Data Presentation: While the study presents extensive data on outcomes across disease groups, there is a need for clearer presentation and interpretation of results. Providing visual aids such as tables or graphs could enhance the readability and accessibility of the findings for readers.

Discussion of Methodological Limitations: The article lacks a comprehensive discussion of the limitations inherent in the methodology employed. Addressing potential biases, data limitations, and other methodological constraints would strengthen the study's validity and assist readers in interpreting the results accurately.

Contextual Analysis: While the study identifies the impact of COVID-19 on hospital care quality, it could benefit from a deeper contextual analysis of the underlying factors influencing these trends. Exploring socioeconomic determinants, healthcare infrastructure, and public health interventions could enrich the interpretation of findings and provide actionable insights for policymakers and healthcare practitioners.

Future Research Directions: The article would benefit from a section dedicated to suggesting avenues for future research. Identifying gaps in knowledge, proposing alternative methodologies, or recommending areas for further investigation would enhance the scholarly contribution of the study and guide future research efforts in this area.

In conclusion, the article "Quality of Hospital Care for WHO's Three Disease Groups Pre-, During, and Post-COVID-19 Pandemic Peak in Thailand" presents a valuable contribution to understanding healthcare quality dynamics during the COVID-19 pandemic. While the study demonstrates strengths in relevance, methodology, and findings, there are opportunities for improvement in data presentation, discussion of methodological limitations, contextual analysis, and identification of future research directions. Addressing these areas would enhance the overall quality and impact of the research.

Reviewer #3: Dear Authors,

The manuscript described hospital quality variables in different periods of COVID-19 pandemic.

I have some comments as follows:

1. Please modify the title to be more precise and attractive, i.e., include some information about diseases in focus and which hospital quality's parameter were measured and reported.

2. Introduction:

- please include more relevant references, esp. profound healthcare disruption from the pandemic (ref.1), and ref. 2. For example, information from CDC or WHO.

- Moreover, there were a lot of published articles about the impact of the pandemics on healthcare quality, including those variables focused on this manuscript. Please elaborate on how would this manuscript benefit the scientific society?

3. Method:

- please use weighted data to analyse the regression, since the sample size of post pandemic period was much smaller than other period.

- what are the effect size of each primary outcome

4. Results:

- please add more information on demographic data in each period. For example, numbers of poverty-related/non-communicable cases in each period.

- regression table: it was interesting that there were significance differences among periods; however, they did not show which period differ from which. From the analytic results presented, it can be that pre-pandemic period gave differences in outcomes than post-pandemic period without the relation to pandemic period.

- 'DM' and 'CVD' are covariate, having both of them in the model can induce bias.

5. Discussion:

- some analysis demonstrated similar levels and trends in during and post-COVID-19 periods.

The authors discussed only the differences during the pandemic, and seemed to neglect the post pandemic's data.

- second page of discussion, line 12-14 -> please reference the statement:

11 Lockdowns may limit people's ability

12 to seek timely healthcare for various conditions, leading to more severe presentations when they

13 eventually seek treatment. More severe cases often require longer hospital stays or hospital

14 mortality.

-second page of discussion,line 26; indicating potential challenges in managing these conditions -> please elaborate on this statement.

- second page of discussion,line 28-29:' The findings for CVD and DM exhibited mixed directions, highlighting the complexity of the pandemic's influence on hospital care across various health issues.' since these diseases can be covariate , it is possible that they manifested similar issues.

- Please revise Academic written English language, particularly in discussion part'

- please add results' implications into the last part of the discussion or in conclusion.

Best regards,

6. PLOS authors have the option to publish the peer review history of their article (what does this mean?). If published, this will include your full peer review and any attached files.

Reviewer #1: No

Reviewer #2: No

Reviewer #3: No

---

## [Author Response · Author response to Decision Letter 0]

4 Aug 2024

We have added the file responses to the reviewers. With the same information. The table of responses in that file may not be put here. So please read that file for more details.

1. Additional requirement 1

Please ensure that your manuscript meets PLOS ONE's style requirements, including those for file naming

Response

We have already re-checked and revised PLOS ONE's style requirements.

2. Additional requirement 2

Thank you for stating the following financial disclosure:

"Research reported in this publication was supported by Fogarty International Center and the National Institute of Allergy and Infectious Diseases of the National Institutes of Health on the project “TB/MDR-TB Research Capacity Building in low and middle-income countries in Southeast Asia” under Awarded Number D43TW009522."

Response

• We have added the statement: "The funders had no role in study design, data collection and analysis, decision to publish, or preparation of the manuscript." in the PLOS ONE submission system.

• We added this amended Role of Funder statement in the cover letter (attached).

3. Additional requirement 3

Thank you for stating the following in the Acknowledgments Section of your manuscript: 

"Thanks to the Fogarty International Center and the National Institute of Allergy and Infectious Diseases of the National Institutes of Health on the project “TB/MDR-TB Research Capacity Building in low and middle-income countries in Southeast Asia” under Awarded Number D43TW009522, for the support."

"Research reported in this publication was supported by Fogarty International Center and the National Institute of Allergy and Infectious Diseases of the National Institutes of Health on the project “TB/MDR-TB Research Capacity Building in low and middle-income countries in Southeast Asia” under Awarded Number D43TW009522."

Response

We revised as suggested in the acknowledgments as follows:

This study was a part of a a post-doctoral trainee of Satiti Palupi at the Department of Epidemiology, Faculty of Medicine, Prince of Songkla University, Hat Yai, Songkhla, Thailand. We also thank the Thai Health Information Portal (THIP) for providing the dataset.

Also, we added amended statements within the cover letter (attached).

4. Additional requirement 4

We note that you have indicated that there are restrictions to data sharing for this study. For studies involving human research participant data or other sensitive data, we encourage authors to share de-identified or anonymized data. However, when data cannot be publicly shared for ethical reasons, we allow authors to make their data sets available upon request. For information on unacceptable data access restrictions, please see http://journals.plos.org/plosone/s/data-availability#loc-unacceptable-data-access-restrictions. 

Response

The study was conducted in accordance with the Declaration of Helsinki, and approved by the Human Research Ethics Committee, Prince of Songkla University (REC number 66-485-18-1). Patient data were encrypted for personalized anonymization according to the Thai Personal Data Protection Act 2019, Thailand (PDPA). All information in the analysis is available in the Supplementary Files. The full data sets of all patients could not be shared according to PDPA.

Data is available to request following the processes described in "https://thip.nbt.or.th/documents/THIP-Manual.pdf " in which detailed procedure of data request and contact information were included. 

5. Additional requirement 5

Please include your full ethics statement in the ‘Methods’ section of your manuscript file. In your statement, please include the full name of the IRB or ethics committee who approved or waived your study, as well as whether or not you obtained informed written or verbal consent. If consent was waived for your study, please include this information in your statement as well.

Response

As suggested in the Materials and Methods section (attached), we have added the information above.

Before revision

In this study, THIP data was accessed between September 5, 2023, and December 16, 2023. The Institutional Ethics Committee of the Faculty of Medicine at Prince of Songkla University in Hat Yai, Thailand, granted ethical approval under reference number REC 66-485-18-1.

After revision

In this study, THIP data was accessed between September 5, 2023, and December 16, 2023. The Institutional Ethics Committee of the Faculty of Medicine at Prince of Songkla University in Hat Yai, Thailand, granted ethical approval under reference number REC 66-485-18-1. Prof. Boonsin Tangtrakulwanich, Chairman of the Human Research Ethics Committee, Faculty of Medicine, Prince of Songkla University, approved the study.

6. Additional requirement 6

 Please modify the title to be more precise and attractive, i.e., include some information about diseases in focus and which hospital quality's parameter were measured and reported.

Response

We changed the title as suggested.

Before revision

Quality of hospital care for WHO's three disease groups pre-, during, and post-COVID-19 pandemic peak in Thailand

After revision

COVID-19's Impact on Hospital Stays, Mortality, and Readmissions for Poverty-Related Diseases, Noncommunicable Diseases, and Injury Groups in Thailand

Feedback to author

Reviewer 1

The study is not novel, and similar papers have been published before.

Response

Thank you for your valuable feedback.

We acknowledge that previous studies have examined the impact of the COVID-19 pandemic on healthcare utilization and outcomes. However, our study makes a distinct contribution by focusing specifically on the Thai healthcare context and analyzing trends in hospital care quality for three distinct disease groups (poverty-related, noncommunicable, and injuries) across the pre-, during-, and post-pandemic peak periods. We believe that these findings have important implications for healthcare planning and resource allocation in Thailand, especially as the country continues to recover from the pandemic.

Reviewer 2

Reviewer #2: The article titled "Quality of Hospital Care for WHO's Three Disease Groups Pre-, During, and Post-COVID-19 Pandemic Peak in Thailand," submitted to the scientific journal Plos One, delves into the critical examination of hospital care quality across different disease groups during various phases of the COVID-19 pandemic. Through analyzing trends in hospital admission data from the Thai Health Information Portal (THIP) database, the study aims to shed light on how the pandemic has affected the quality of care for poverty-related, noncommunicable, and injury-related diseases in Thailand.

Positive Points:

Relevance and Timeliness: The study addresses a timely and significant issue, providing insights into how the COVID-19 pandemic has impacted the quality of hospital care across different disease groups. This research is particularly valuable given the ongoing global healthcare crisis and the need to understand its implications on healthcare delivery systems.

Comprehensive Methodology: The utilization of existing hospital admission data from the THIP database covering a substantial period from 2017 to 2022 lends credibility to the study. Categorizing disease groups using the International Classification of Diseases (ICD)-10 and analyzing outcomes such as prolonged length of stay (LOS), hospital mortality, and readmission pre-, during, and post-COVID-19 pandemic peak offers a comprehensive view of the situation.

Insightful Findings: The study's findings provide valuable insights into the impact of the pandemic on different disease groups. By examining metrics such as prolonged LOS, hospital mortality rates, and readmissions, the research highlights the varying degrees of impact across poverty-related, noncommunicable, and injury-related diseases, contributing to a deeper understanding of healthcare dynamics during the pandemic.

Opportunities for Improvement:

Clarity in Data Presentation: While the study presents extensive data on outcomes across disease groups, there is a need for clearer presentation and interpretation of results. Providing visual aids such as tables or graphs could enhance the readability and accessibility of the findings for readers.

Discussion of Methodological Limitations: The article lacks a comprehensive discussion of the limitations inherent in the methodology employed. Addressing potential biases, data limitations, and other methodological constraints would strengthen the study's validity and assist readers in interpreting the results accurately.

Contextual Analysis: While the study identifies the impact of COVID-19 on hospital care quality, it could benefit from a deeper contextual analysis of the underlying factors influencing these trends. Exploring socioeconomic determinants, healthcare infrastructure, and public health interventions could enrich the interpretation of findings and provide actionable insights for policymakers and healthcare practitioners.

Future Research Directions: The article would benefit from a section dedicated to suggesting avenues for future research. Identifying gaps in knowledge, proposing alternative methodologies, or recommending areas for further investigation would enhance the scholarly contribution of the study and guide future research efforts in this area.

In conclusion, the article "Quality of Hospital Care for WHO's Three Disease Groups Pre-, During, and Post-COVID-19 Pandemic Peak in Thailand" presents a valuable contribution to understanding healthcare quality dynamics during the COVID-19 pandemic. While the study demonstrates strengths in relevance, methodology, and findings, there are opportunities for improvement in data presentation, discussion of methodological limitations, contextual analysis, and identification of future research directions. Addressing these areas would enhance the overall quality and impact of the research.

Response

Thank you for your insightful comments and positive feedback on our manuscript. We have already added sentences and revised them for improvement as suggested.

Clarity in Data Presentation: 

While the study presents extensive data on outcomes across disease groups, there is a need for clearer presentation and interpretation of results. Providing visual aids such as tables or graphs could enhance the readability and accessibility of the findings for readers.

RESPONSE:

For Clarity in Data Presentation, we have enhanced the presentation of results by incorporating tables and graphs to provide a clearer and more accessible overview of the findings. Specifically, we have added a table summarizing the direction of effects and consistent findings among the three major disease groups (Table 7). Additionally, we have included supplementary tables (S1, S2, and S3 Tables) presenting the results of logistic regression analyses, further clarifying the statistical associations observed.

Discussion of Methodological Limitations: 

The article lacks a comprehensive discussion of the limitations inherent in the methodology employed. Addressing potential biases, data limitations, and other methodological constraints would strengthen the study's validity and assist readers in interpreting the results accurately.

RESPONSE

For the Discussion of Methodological Limitations, we already revised the Discussion section in paragraph 15

Before revision

 The limitations of this study were that it did not conclude all the diseases are there hospitalized, and the data utilized may not have comprehensively captured the broader health outcomes of the entire population, as the study primarily relies on hospital-based metrics and lack of information level of complication. That limitation emphasizes the need for future research to adopt a more holistic approach, incorporating community-level health indicators and qualitative assessments to better understand the comprehensive implications of healthcare changes during and post-COVID-19 pandemic peak.

After revision:

 This study has several limitations. First, it relied on hospital admission data, which might not fully capture the broader health outcomes of the entire population. Second, the data lacks information on disease severity, which could be a potential confounder in the analysis. Third, the use of relative weight (RW) as a surrogate measure of disease severity has its limitations, as it may not fully reflect the complexity and heterogeneity of individual cases. Finally, the study is observational. These limitations emphasize the need for future research to adopt a more holistic approach, incorporating community-level health indicators and qualitative assessments to better understand the comprehensive implications of healthcare changes during and post-COVID-19 pandemic peak.

Contextual Analysis: 

While the study identifies the impact of COVID-19 on hospital care quality, it could benefit from a deeper contextual analysis of the underlying factors influencing these trends. Exploring socioeconomic determinants, healthcare infrastructure, and public health interventions could enrich the interpretation of findings and provide actionable insights for policymakers and healthcare practitioners.

RESPONSE:

Our manuscript processes data from the THIP database or patients being hospitalized in Thailand. The data we used did not include information about determinants of socioeconomic status such as income, education, or occupation. However, this manuscript could provide a glimpse of information regarding SES from the group of diseases related to poverty-related diseases.

 If we discuss healthcare infrastructure and public health intervention, this is a limitation of this manuscript.

Future Research Directions: 

The article would benefit from a section dedicated to suggesting avenues for future research. Identifying gaps in knowledge, proposing alternative methodologies, or recommendin

---

## [Decision Letter · Decision Letter 1]

26 Aug 2024

COVID-19’s Impact on Hospital Stays, Mortality, and Readmissions for Poverty-related Diseases, Noncommunicable Diseases, and Injury Groups

in Thailand

PONE-D-24-06438R1

Dear Dr. Chongsuvivatwong,

We’re pleased to inform you that your manuscript has been judged scientifically suitable for publication and will be formally accepted for publication once it meets all outstanding technical requirements.

Kind regards,

Worradorn Phairuang, Ph.D.

Academic Editor

PLOS ONE

Additional Editor Comments (optional):

Reviewers' comments:

Reviewer's Responses to Questions

**Comments to the Author**

1. If the authors have adequately addressed your comments raised in a previous round of review and you feel that this manuscript is now acceptable for publication, you may indicate that here to bypass the “Comments to the Author” section, enter your conflict of interest statement in the “Confidential to Editor” section, and submit your "Accept" recommendation.

Reviewer #2: All comments have been addressed

2. Is the manuscript technically sound, and do the data support the conclusions?

Reviewer #2: Yes

3. Has the statistical analysis been performed appropriately and rigorously? 

Reviewer #2: Yes

4. Have the authors made all data underlying the findings in their manuscript fully available?

Reviewer #2: Yes

5. Is the manuscript presented in an intelligible fashion and written in standard English?

Reviewer #2: Yes

6. Review Comments to the Author

Reviewer #2: The authors have performed the suggested suggestions. The manuscript can be accepted for publication. Congratulations!

7. PLOS authors have the option to publish the peer review history of their article (what does this mean?). If published, this will include your full peer review and any attached files.

Reviewer #2: No

---

## [Editor Report · Acceptance letter]

1 Sep 2024

PONE-D-24-06438R1 

PLOS ONE

Dear Dr. Chongsuvivatwong, 

I'm pleased to inform you that your manuscript has been deemed suitable for publication in PLOS ONE. Congratulations! Your manuscript is now being handed over to our production team.

Kind regards, 

on behalf of

Dr. Worradorn Phairuang 

Academic Editor

PLOS ONE